# Biologically informed cortical models predict optogenetic perturbations

**Christos Sourmpis[1,2], Carl CH Petersen[2], Wulfram Gerstner[1], Guillaume Bellec[1,3]\***

[1]Laboratory of Computational Neuroscience, Brain Mind Institute, School of Computer and Communication Sciences and School of Life Sciences, École Polytechnique Fédérale de Lausanne (EPFL), Lausanne, Switzerland; [2]Laboratory of Sensory Processing, Brain Mind Institute, School of Life Sciences, École Polytechnique Fédérale de Lausanne (EPFL), Lausanne, Switzerland; [3]Machine Learning Research Unit, Technical University of Vienna (TU Wien), Vienna, Austria

## eLife Assessment

This **important** study demonstrates the significance of incorporating biological constraints in training neural networks to develop models that make accurate predictions under novel conditions. By comparing standard sigmoid recurrent neural networks (RNNs) with biologically constrained RNNs, the manuscript offers **compelling** evidence that biologically grounded inductive biases enhance generalization to perturbed conditions. This manuscript will appeal to a wide audience in systems and computational neuroscience.

**\*For correspondence:**
guillaume.bellec@epfl.ch

**Abstract** A recurrent neural network fitted to large electrophysiological datasets may help us understand the chain of cortical information transmission. In particular, successful network reconstruction methods should enable a model to predict the response to optogenetic perturbations. We test recurrent neural networks (RNNs) fitted to electrophysiological datasets on unseen optogenetic interventions and measure that generic RNNs used predominantly in the field generalize poorly on these perturbations. Our alternative RNN model adds biologically informed inductive biases like structured connectivity of excitatory and inhibitory neurons and spiking neuron dynamics. We measure that some biological inductive biases improve the model prediction on perturbed trials in a simulated dataset and a dataset recorded in mice in vivo. Furthermore, we show in theory and simulations that gradients of the fitted RNN can be used to target micro-perturbations in the recorded circuits and discuss the potential utility to bias an animal's behavior and study cortical circuit mechanisms.

## Introduction

A fundamental question in neuroscience is how cortical circuit mechanisms drive perception and behavior. To tackle this question, experimental neuroscientists have been collecting large-scale electrophysiology datasets under reproducible experimental settings (*Siegle et al., 2021*; *Esmaeili et al., 2021*; *Urai et al., 2022*; *Benson et al., 2023*). However, neuroscience lacks data-grounded modeling approaches to generate and test hypotheses on the causal role of neuronal and circuit-level mechanisms. To leverage the high information density of contemporary recordings, we need both (1) modeling approaches that scale well with data, and (2) metrics to quantify when the models provide a plausible mechanism for the observed phenomena.

Biophysical simulations have been crucial for our understanding of single-cell mechanisms (*Hodgkin, 1958*), and have been used to describe interactions across cortical layers, columns, and

areas (*Markram et al., 2015*; *Billeh et al., 2020*; *Isbister et al., 2023*; *Chen et al., 2022*; *Rimehaug et al., 2023*; *Fraile et al., 2023*; *Spieler et al., 2023*). A promising approach to constrain models to electrophysiological data lies in the optimization of the simulation parameters by gradient descent. These methods were successful in quantitatively classifying functional cell types (*Pozzorini et al., 2015*; *Teeter et al., 2018*), and modeling micro-circuit interactions (*Pillow et al., 2008*; *Deny et al., 2017*; *Mahuas et al., 2020*). To bridge the gap from single neurons or small retinal networks to cortical recordings in vivo, recent studies made substantial progress towards data-constrained recurrent neural network (RNN) models (*Perich et al., 2020*; *Bellec et al., 2021*; *Arthur et al., 2023*; *Valente et al., 2022*; *Kim et al., 2023*; *Shai et al., 2023*; *Sourmpis et al., 2023*; *Pals et al., 2024*). In this line of work, neurons in the RNN are mapped one-to-one to recorded cells and optimized by gradient descent to predict recorded activity at large scale.

An important question is whether these data-constrained RNNs can reveal a truthful mechanism of neuronal activity and behavior. By construction, the RNNs can generate brain-like network activity, but how can we measure whether the reconstructed network faithfully represents the biophysical mechanism? To answer this question, we submit a range of RNN reconstruction methods to a difficult *perturbation test*: we measure the similarity of the network response to unseen perturbations in the RNN and the recorded biological circuit.

Optogenetics is a powerful tool to induce precise causal perturbations in vivo (*Esmaeili et al., 2021*; *Guo et al., 2014*). It involves the expression of light-sensitive ion channels (*Boyden et al., 2005*), such as channelrhodopsins, in specific populations of neurons (e.g. excitatory/pyramidal or inhibitory/parvalbumin-expressing). In this paper, we use datasets including both dense electrophysiological recordings and optogenetic perturbations to evaluate RNN reconstruction methods. Since the neurons in our RNNs are mapped one-to-one to the recorded cells, we can model optogenetic perturbations targeting the same cell types and areas as done in vivo. Yet, we observe that the similarity between the simulated and recorded perturbations varies greatly depending on the reconstruction methods.

Most prominently, we study two opposite types of RNN specifications. First, as a control model, we consider a traditional sigmoidal RNN (σRNN) which is arguably the most common choice for contemporary data-constrained RNNs (*Perich et al., 2020*; *Arthur et al., 2023*; *Pals et al., 2024*); and second, we develop a model with biologically informed inductive biases (bioRNN): (1) neuronal dynamics follow a simplified spiking neuron model, and (2) neurons associated with fast-spiking inhibitory cells have short-distance inhibitory projections (other neurons are excitatory with both local and long-range interareal connectivity). Following *Neftci et al., 2019*; *Bellec et al., 2018b*; *Bellec et al., 2021*; *Sourmpis et al., 2023*, we adapt gradient descent techniques to optimize the bioRNN parameters of neurons and synapses to explain the recorded neural activity and behavior.

Strikingly, we find that the bioRNN is more robust to perturbations than the σRNN. This is nontrivial because it is in direct contradiction with other metrics often used in the field: the σRNN simulation achieves higher similarity with unseen recorded trials before perturbation, but lower than the bioRNN on perturbed trials. This contradiction is confirmed both on synthetic and in vivo datasets. To analyze this result, we submit a spectrum of intermediate bioRNN models to the same *perturbation tests* and identify two bioRNN model features that are most important to improve robustness to perturbation: (1) Dale's law (the cell type constrains the sign of the connections; *Eccles, 1976*), and (2) local-only inhibition (inhibitory neurons do not project to other cortical areas). In contrast, other model features are penalizing or do not improve significantly the prediction of the optogenetically perturbed response in this out-of-distribution fashion. It indicates that perturbation tests can validate biophysical modeling strategies in data-constrained deep learning models of neural mechanisms.

Beyond the optogenetic area inactivation available in the in vivo dataset, we investigate how perturbation-robust RNNs could enable targeted optogenetic protocols for the discovery of detailed neuronal circuit mechanisms in future experiments. Targeted causal interventions will become decisive in studying smaller circuit mechanisms. Acute optogenetic inactivations of genetically defined laminar subpopulations were used to characterize the causal role of specific neurons in the sensory motor pathways (*Tamura et al., 2025*; *Wyart et al., 2025*), and upcoming technology will make these experiments easier (*Lakunina et al., 2025*). To illustrate how RNN reconstruction can help to target neuronal stimulation, we consider micro-perturbations (μ-perturbation) targeting dozens of neurons in a small time window. Inspired by recent read-write all-optical setups (*Packer et al., 2015*), we imagine a

model-informed μ-perturbation protocol, where neurons are targeted based on their functional rather than genetic properties. While previous work has used linear models to produce targeted stimulations (*Wyart et al., 2025*; *Minai et al., 2024*), we show that back-propagated gradients of perturbation-robust RNNs provide a sensitivity map to predict the effect of μ-perturbations. Concretely, in a closed-loop experimental setup in silicon, we can use RNN gradients to target a μ-perturbation and change the movement in a simulated mouse. The gradients are used to identify the few neurons having the strongest causal effect on behavior. Conceptually, it means that our RNN reconstructions enable an estimation of 'circuit gradients', bringing numerical and theoretical concepts from deep learning (*LeCun et al., 2015*; *Richards and Kording, 2023*) to study biological network computation.

## Results

## Reconstructed networks: biological inductive biases strengthen robustness to perturbations

### Synthetic dataset for challenging causal inference

We build a toy synthetic dataset to formalize how we intend to reverse engineer the mechanism of a recorded circuit using optogenetic perturbations and RNN reconstruction methods. It also serves as the first dataset to evaluate our network reconstruction methods. This toy example represents a simplified version of large-scale cortical recordings from multiple brain areas during a low-dimensional instructed behavior (*Steinmetz et al., 2019*; *Esmaeili et al., 2021*), similarly to the in vivo dataset of a GO/No-Go task *Esmaeili et al., 2021* analyzed in the next section. Let's consider two areas $A$ and $B$ which are either transiently active together ('hit trial' occurring with frequency $p$) or quiescent together ('miss trial' occurring with probability $1 - p$). Since the two areas are active or inactive together, it is hard to infer if they are connected in a feedforward or recurrent fashion. In Methods (Mathematical toy model of the difficult causal inference between H1 and H2), we describe a theoretical example where it is impossible to decide between opposing mechanistic hypotheses (feedforward or recurrent) when recording only the macroscopic activations of areas $A$ and $B$. In this case, performing optogenetic inactivation of one area is decisive to distinguish between the feedforward or recurrent hypothesis.

To generate artificial spike train recordings that capture this problem, we design two reference circuits (RefCircs) from which we can record the spike trains. Each RefCirc consists of two populations of 250 spiking neurons (80% are excitatory) representing areas $A$ and $B$. To highlight the importance of optogenetic perturbations as in the Methods (Mathematical toy model of the difficult causal inference between H1 and H2), the first circuit RefCirc1 is feedforward and the second RefCirc2 is recurrent: RefCirc1 (and not RefCirc2) has strictly zero feedback connections from $B$ to $A$. Yet, the two RefCircs are almost identical without optogenetic perturbations: each neuron in RefCirc1 has been constructed to have an almost identical trial-averaged activity as the corresponding neuron in RefCirc2; and in response to a stimulus, the circuits display a similar bi-modal hit-or-miss response with a hit trial frequency $p \approx 50\%$. We consider that a trial is a hit if area $A$ is active, if the averaged firing rate is above 8 Hz. Note that defining a 'hit' trial based on area $A$ is equivalent to saying that both areas need to be active during unperturbed trials with this dataset. But excluding $B$ in this definition avoids that the hit rate is trivially impacted when manipulating the activity of $B$ with optogenetic perturbations. To simulate optogenetic inactivations of an area in the RefCircs, we inject a transient current into the inhibitory neurons, modeling the opening of light-sensitive ion channels. Symmetrically, an optogenetic activation is simulated as a positive current injected into excitatory cells. *Figure 1—figure supplement 1* shows that optogenetic perturbations in area B reflect the presence or absence of feedback connections which differ in RefCirc 1 and 2. Methods section Reference circuits for hypotheses 1 and 2 provides more details on the construction of the artificial circuits. Our *perturbation test* will consist of the comparison of optogenetic perturbations in the reconstructed RNN and in their references RefCirc1 and 2, without retraining the RNN on these perturbations.

### Network reconstruction methodology (synthetic dataset)

To reconstruct the recorded circuits with an RNN, we record activity from the spiking RefCircs and optimize the parameters of an RNN to generate highly similar network activity. The whole reconstruction method is summarized graphically in panel A of *Figure 1*. In the simplest cases, the RNN is specified as a sigmoidal network model (*Rosenblatt, 1960*; *Elman, 1990*): σRNN1 and σRNN2 are optimized

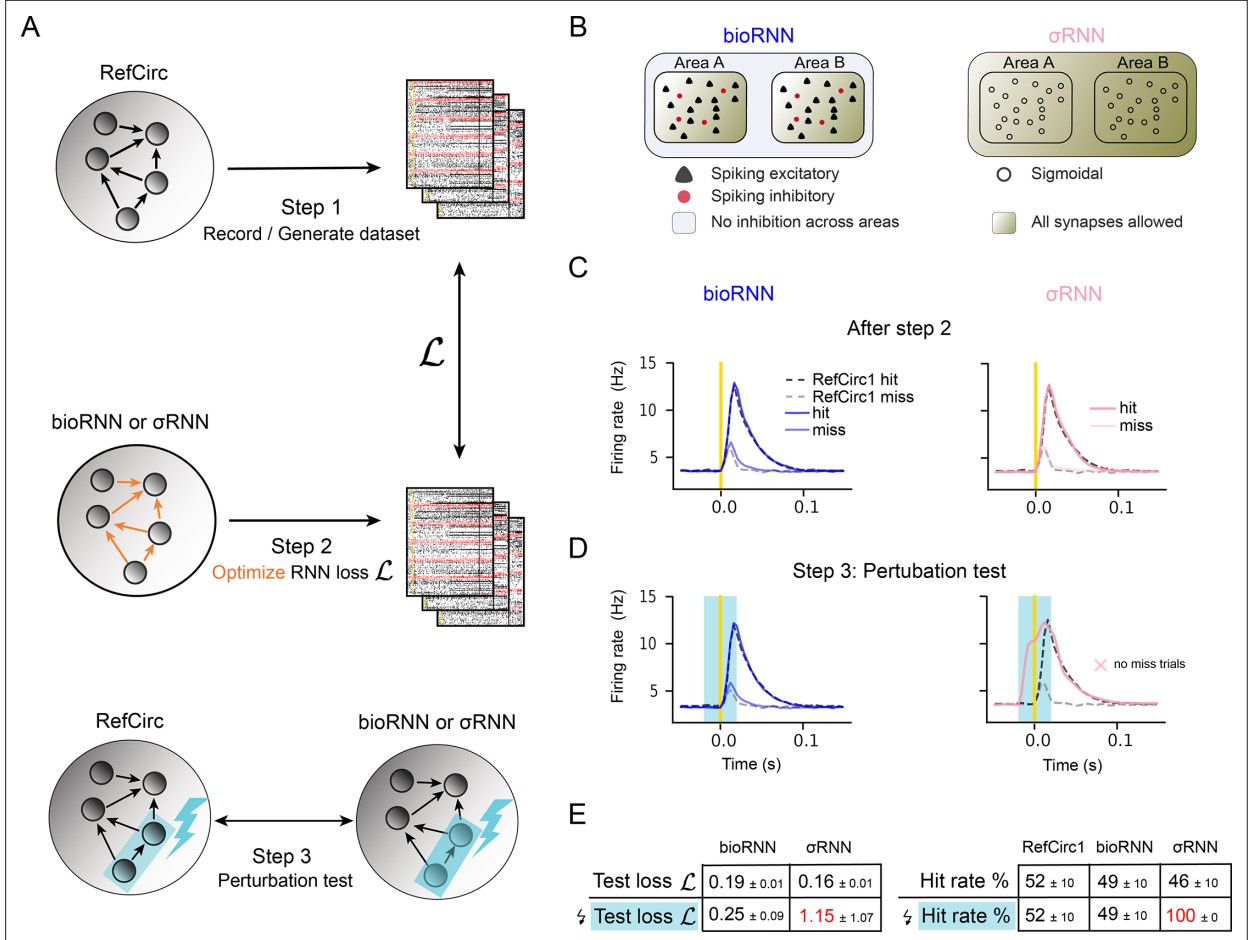

**Figure 1.** Network reconstruction and perturbation tests. (**A**) The three steps to reconstruct the reference circuit (RefCirc) using a biologically informed RNN (bioRNN) or a sigmoidal RNN (σRNN) and evaluate the reconstruction based on perturbation tests. (**B**) Summary of the differences between a bioRNN and a σRNN. (**C**) Trial-averaged activity of area *A* of the two circuits during hit (black-dashed: RefCirc1; blue: bioRNN1; pink: σRNN1) and miss (grey-dashed: RefCirc1; light blue: bioRNN1; light pink: σRNN1) trials. All models display a hit rate of $p \approx 50\%$. (**D**) Same as **C** during inactivation of area *B*. $\Delta p^{\mathcal{D}} = 0$ is the recorded change of hit rate for the feedforward circuit RefCirc1, so a successful reconstruction achieves $\hat{p} \approx 0\%$. (**E**) Quantitative results on perturbation tests showing that σRNN achieves the lowest loss function on the unperturbed test trials, but only the bioRNN retains an accurate fit to the perturbed trials.

The online version of this article includes the following figure supplement(s) for figure 1:

**Figure supplement 1.** Modeling 'optogenetic' perturbations.

to reproduce the recording from RefCirc1 and RefCirc2, respectively. In this synthetic dataset, the reconstructed σRNNs have the same size as the RefCircs (500 neurons) and sigmoidal neurons are mapped one-to-one with RefCirc neurons (20% are mapped to inhibitory RefCirc neurons). They are initialized with all-to-all connectivity and are therefore blind to the structural difference of the RefCirc1 and 2 (feedforward or recurrent). From each of the two RefCircs, we store 2000 trials of simultaneous spike train recordings of all 500 neurons (step 1 in *Figure 1A*). Half of the trials are used as a training set and will be the basis for our data-driven RNN optimization. The second half of the recorded trials forms the testing set and is used to evaluate the quality of the reconstruction before perturbations.

We optimize the synaptic 'weights' of the σRNN to minimize the difference between its activity and that of the RefCirc (step 2 in *Figure 1A*, see Methods). The optimization combines three loss functions defined mathematically in Methods (Optimization and loss function): (i) the neuron-specific loss function $\mathcal{L}_{\mathrm{neuron}}$ is the mean-square error of the *trial-averaged* neural activity (e.g. the PSTH) between σRNN and RefCirc neurons. (ii) To account for fluctuations of the single-trial network activity, we use a trial-specific loss function $\mathcal{L}_{\mathrm{trial}}$, which is the distance between the distribution of single trial

**Table 1.** BioRNN is more robust to optogenetic perturbations than σRNN.

The table reports the trial matching (TM) loss $\mathcal{L}_{\text{trial}}$ on test trials; it measures the distance between the distributions of single trial network dynamics *Sourmpis et al., 2023* in area A when stimulating area $B$. Column 'no light' indicates values on the unperturbed test trials, and 'light' the perturbation trials. ± indicates the 95% confidence interval, best values are shown in bold and major failure with distance above 0.5 is in red.

| | RefCirc1 vs. RNN1 | | RefCirc2 vs. RNN2 | |
| --- | --- | --- | --- | --- |
| | no light | light | no light | light |
| bioRNN | 0.19 ± 0.01 | 0.25 ± 0.09 | 0.18 ± 0.01 | **0.28 ± 0.13** |
| σRNN | **0.16 ± 0.01** | 1.15 ± 1.07 | **0.17 ± 0.01** | 1.22 ± 0.64 |
| No sparsity | 0.20 ± 0.01 | 1.37 ± 1.42 | 0.19 ± 0.01 | 0.19 ± 0.13 |
| Non-local inhibition | 0.20 ± 0.02 | 0.54 ± 0.42 | 0.18 ± 0.01 | 1.19 ± 0.91 |
| No Dale's law | 0.18 ± 0.01 | 0.86 ± 0.23 | 0.18 ± 0.01 | 2.21 ± 1.60 |
| No spike | 0.17 ± 0.00 | **0.19 ± 0.04** | 0.18 ± 0.00 | 0.46 ± 0.19 |
| No Trial Matching (TM) | 0.33 ± 0.01 | 0.44 ± 0.19 | 0.35 ± 0.03 | 0.44 ± 0.09 |

population-averaged activity of σRNN and RefCirc (see *Sourmpis et al., 2023*). (iii) Finally, we add a regularization loss function $\mathcal{L}_{\text{reg}}$ to penalize unnecessarily large weights.

We also developed a biologically informed RNN model (bioRNN) for which we have designed a successful optimization technique. The main differences between σRNNs and bioRNNs consist of the following biological inductive biases. Firstly, the bioRNN neuron model follows a simplified leaky integrate and fire dynamics (see Methods, Neuron and jaw movement model) yielding strictly binary spiking activity. Secondly, we constrain the recurrent weight matrix to describe cell-type-specific connectivity constraints: following Dale's law, neurons have either non-negative or non-positive outgoing connections; moreover, since cortical inhibitory neurons rarely project across areas, we assume that inhibitory neurons project only locally within the same area. Thirdly, we add a term to the regularization loss $\mathcal{L}_{\text{reg}}$ to implement the prior knowledge that cross-area connections are more sparse than within an area. Adding these biological features into the model requires an adapted gradient descent algorithm and matrix initialization strategies (Methods, Optimization and loss function). The reconstruction method with σRNNs and bioRNNs is otherwise identical: the models have the same size and are optimized on the same data for the same number of steps and using the same loss functions. The two models bioRNN1 and bioRNN2 are optimized to explain recordings from RefCirc1 and Refcirc2, respectively. Importantly, the structural difference between RefCirc1 (feedforward) and RefCirc2 (feedback) is assumed to be unknown during parameter optimization: at initialization, excitatory neurons in bioRNN1 or bioRNN2 project to any neuron in the network with transmission efficacies (aka as synaptic weights) initialized randomly.

After parameter optimization, we have four models, σRNN1, σRNN2, bioRNN1, and bioRNN2, that we call 'reconstructed' models. To validate the reconstructed models, we verify that the network trajectories closely match the data on the test set in terms of (i) the 'behavioral' hit-trial frequency, (ii) the peristimulus time histogram (PSTH) mean-square error of single neurons as evaluated by $\mathcal{L}_{\text{neuron}}$, and (iii) the distance between single-trial network dynamics as evaluated by $\mathcal{L}_{\text{trial}}$ (see *Figure 2—figure supplement 1* and *Table 1*). At first sight, the σRNN displays a better data fitting when comparing with the non-perturbed trials of the testing set: $\mathcal{L}_{\text{trial}}$ is for instance, lower with σRNN (see *Table 1*). This is expected considering that the optimization of bioRNNs is less flexible and numerically efficient because of the sign-constrained weight matrix and the imperfect surrogate gradient approximation through spiking activity. However, the two bioRNNs are drastically more robust when evaluating the models with *perturbation tests*.

## Perturbation test

To test which of the reconstructed RNNs capture the causal mechanisms of the RefCircs, we simulate optogenetic activations and inactivations of area $B$ (step 3 in *Figure 1A*). We first compare the change of hit probability after perturbations in the reconstructed RNN ($\Delta \hat{p}$) and recorded in RefCirc ($\Delta p^{\mathcal{D}}$) in *Figure 2*. For the σRNN, the activation or inactivation of area $B$ changes drastically the peak

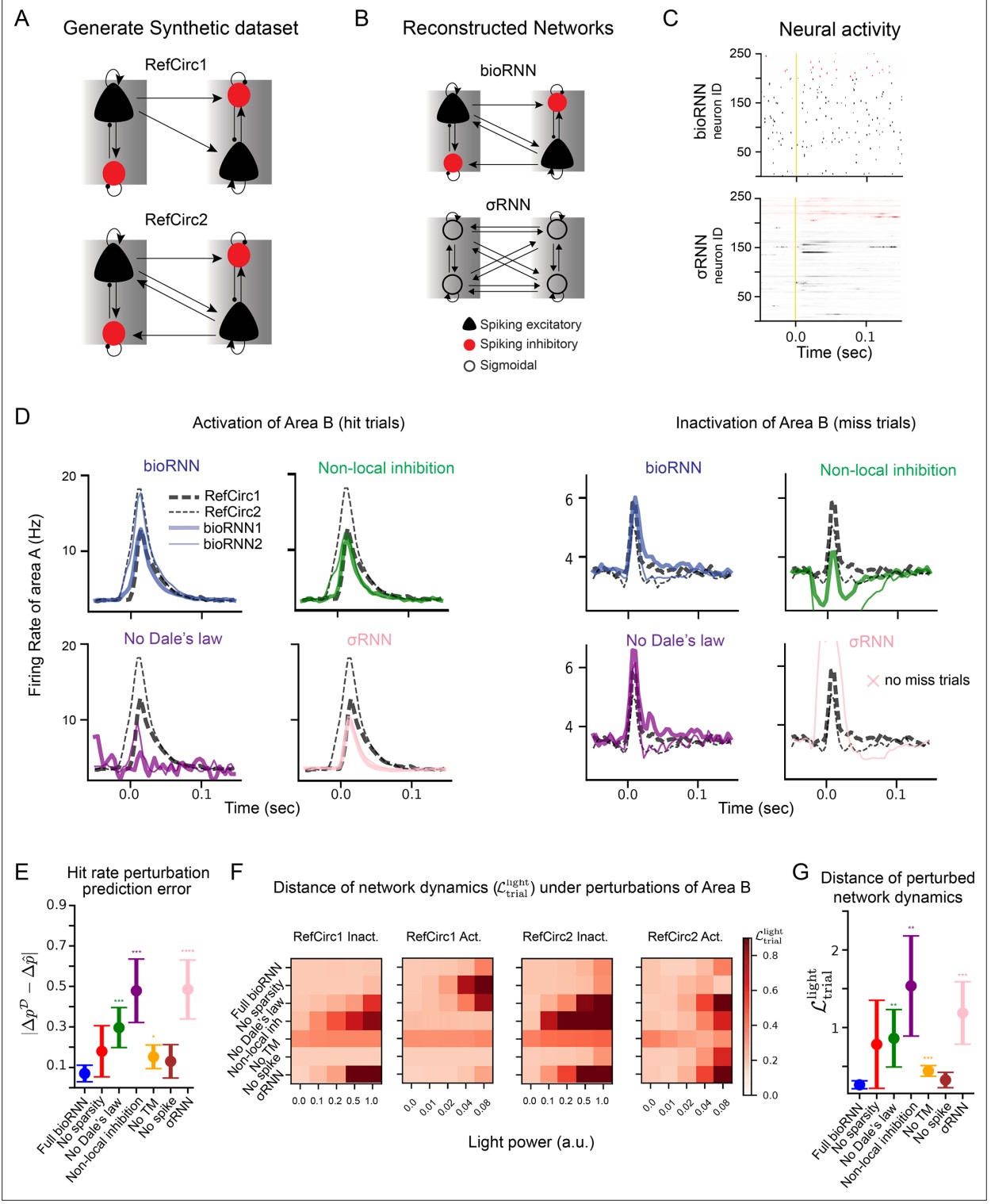

**Figure 2.** Reconstruction of network mechanisms. (**A**) RefCirc1 is feedforward and RefCirc2 is recurrent. (**B**) The fitted RNNs are blind to the structural difference of RefCirc1 and 2 and must infer this from the spiking data. (**C**) Raster plot showing an example trial of the bioRNN and σRNN models, neurons in red are mapped to inhibitory neurons. (**D**) To study which model feature matters, bioRNN variants are defined by removing one of the features, for instance 'No Dale's law' refers to a bioRNN without weight sign constraints. Trial-averaged activity in area $A$ under activation/inactivation of area $B$. All the RNNs are tested with the same reference circuit and training data (No spike and No Sparsity models are shown in *Figure 2—figure supplement 3*). (**E**) Error between the change of hit probability after perturbations in the RNN $\Delta\hat{p}$ and in the RefCirc $\Delta p^{\mathcal{D}}$, the whiskers indicate the 95% confidence interval. (**F**) The distance of network dynamics $\mathcal{L}_{\text{trial}}^{\text{light}}$ between each RNN and RefCirc (horizontal axis: light power in arbitrary units).

*Figure 2 continued on next page*

*Figure 2 continued*

(**G**) Same quantity as *D* but averaged for each RNN under the strongest light power condition (averaging activations and inactivations of area *B*), the whiskers indicate the 95% confidence interval. Statistical significance in comparison with bioRNN is computed with t-test using the mean over multiple network initializations and is indicated with 0–4 stars corresponding to p-values thresholds: 0.05, $10^{-2}$, $10^{-3}$, and $10^{-4}$.

The online version of this article includes the following figure supplement(s) for figure 2:

**Figure supplement 1.** Fitting Reconstructed networks to the synthetic dataset.

**Figure supplement 2.** Picking the sparsity level.

**Figure supplement 3.** Trial-averaged traces across RNN variants.

**Figure supplement 4.** Hit frequency prediction error I$\Delta p^{\mathcal{D}} - \Delta \hat{p}$I as in *Figure 2E*.

---

firing rate in area *A*: all trials become a hit during inactivation of area *B*. This drastic increase in hit rate is not consistent with the reference where the effect of the optogenetic inactivations is mild: the distribution of network responses remains bi-modal (hit versus miss) with only a moderate change of hit frequency for RefCirc2 $\Delta p^{\mathcal{D}} = -3\%$. For RefCirc1, we even expect $\Delta p^{\mathcal{D}} = 0\%$ by design because of the absence of feedback connections from *B* to *A*. In contrast, the bioRNN models capture these changes more accurately (see *Figures 1 and 2*). Quantitative results are summarized in *Figure 2E*, the error of hit probability changes I$\Delta p^{\mathcal{D}} - \Delta \hat{p}$I is 7% with bioRNNs when averaged over all conditions (bioRNN1 and bioRNN2, with optogenetic inactivations and activations). The corresponding error is 48.5% on average for σRNNs. In this sense, we argue that the bioRNN provides a better prediction of the perturbed hit frequency than the σRNN. We also performed spike train recordings in the area that is not directly targeted by the light to compare the perturbed network dynamics in the fitted RNNs and the RefCirc. The perturbed dynamics are displayed in *Figure 2D*. The quantity $\mathcal{L}_{\text{trial}}^{\text{light}}$ is a distance between the network dynamics (RNN versus reference) and is reported in *Figure 2D–E* and *Table 1*. Again, the perturbed dynamics of the bioRNN are more similar to those of the reference circuits $\mathcal{L}_{\text{trial}}^{\text{light}} = 0.26$, than with the σRNN $\mathcal{L}_{\text{trial}}^{\text{light}} = 1.19$ (t-test p-value is 0.0003).

To analyze which features of bioRNN explain this robustness to perturbation, we then derive a family of models where only one feature of the reconstruction is omitted. Namely, the 'No Dale's law' model does not have excitatory and inhibitory weight constraints, the 'Non-local inhibition' model allows inhibitory neurons to project outside of their areas, the 'No Spike' model replaces the spiking dynamics with a sigmoidal neuron model, and the 'No Sparsity' model omits the cross-area sparsity penalty in $\mathcal{L}_{\text{reg}}$. Omitting all these features in bioRNN would be equivalent to using a σRNN. The accuracy metrics on the testing sets before perturbation are reported for all RNN variants in *Figure 2E and G*. For reference, we also include the model 'No TM' (trial-matching), which omits the loss function $\mathcal{L}_{\text{trial}}$ during training.

The strongest effect measured with this analysis is that Dale's law and local inhibition explain most of the improved robustness of bioRNNs. This is visible in *Figure 2* as the perturbed trajectories of 'No Dale's law' and 'Non-local inhibition' are most distant from the reference in *Figure 2D*. This is confirmed numerically where both the hit-rate error and the distance of network dynamics increase the most when lifting these constraints (*Figure 2E–G* and *Table 1*). We explain this result as follows: the mono-synaptic effect of a cell stimulated by the light is always correct in bioRNN (according to Dale's law and inhibition locality), but often wrong in the alternative models (see *Figure 2A*). For instance, a simple explanation may justify the failure of the 'Non-local inhibition' model: the stimulation of inhibitory neurons in *B* induces (via the erroneous mono-synaptic inhibition) a reduction in the baseline activity in area *A* (see the green trace during inactivation in *Figure 2D*). More generally for *perturbation testing*, we speculate that these features are measured to be important here because they are central to the biophysical nature of the perturbation considered: optogenetic perturbation targets specific cell types, and these features incorporate biophysical connectivity priors that are hard to infer entirely from the unperturbed data.

Not all the biological features that we implemented in bioRNN made comparable improvements in the prediction of optogenetic perturbations. We implemented simple spiking neuron dynamics and fitted the spiking network as any other RNN using surrogate gradients (*Neftci et al., 2019*) as in *Bellec et al., 2021*; *Sourmpis et al., 2023*. On perturbed data, the spiking bioRNN achieves slightly better performance than its 'No spike' variant, but without significant margins, t-test p-value is 0.31 for $\mathcal{L}_{\text{trial}}^{\text{light}}$ (see *Table 1* and *Figure 2E–G*). We speculate that simulating spikes is not advantageous

here, because optogenetic perturbations are relatively broad in space and time, and it might become more relevant for other perturbation experiments where precise timing matters or at a microcircuit level. Similarly, the sparse connectivity regularization did not yield a significant improvement on the perturbation tests (see *Table 1*).

Besides predicting the response to optogenetic perturbations, we wondered if we could recover the connectivity structure of the recorded circuit. Our method would not be appropriate to recover individual synaptic connections, but we tested whether the fitted RNNs reflected the optogenetic signature of the structural difference between the 'feedforward' RefCirc1 and the 'recurrent' RefCirc2. Our criteria are here qualitative: the early increase in the PSTH response in area $A$ characteristic of mono-synaptic feedback from area $B$ should not exist for RefCirc1 (see *Figure 2* and *Figure 2—figure supplement 3A*). To reveal this difference in the fitted bioRNN1 and bioRNN2 models, not only Dale's law and local inhibition are necessary, but also spiking dynamics and sparsity appear to be helpful. For instance, the erroneous early onset on the perturbed trial in area A for the 'No Sparsity' model is corrected with the sparsity prior (*Figure 2—figure supplement 3*, red versus blue curves). Yet, these are subtle qualitative results that are likely to be less impactful and reproducible than the clear qualitative improvement obtained with perturbation testing when modeling cell-type connectivity. Indeed, we will see in the next section that we obtain consistent qualitative perturbation testing results on the larger in vivo dataset.

## Predicting perturbations on in vivo electrophysiology data

To test whether our reconstruction method with biological inductive biases can predict optogenetic perturbations in large-scale recordings, we used the dataset from *Esmaeili et al., 2021*. In this study, water-deprived mice were trained to perform a whisker tactile detection task. In 50% of the trials (Go trials), a whisker is deflected, followed by a 1 s delay, after which an auditory cue signals that the mice can lick a water spout to receive a water reward. In the other 50% of trials (No-Go trials), no whisker deflection occurs, and licking after the auditory cue results in a penalty with an extended time-out period. While the mice performed the task, experimenters recorded 6,182 units from 12 areas across 18 mice. Using this dataset, we focused on the six most relevant areas for executing this task (*Esmaeili et al., 2021*). From each area, we randomly selected 250 neurons (200 putative excitatory and 50 putative inhibitory), which correspond to 1500 neurons in total. These areas, all shown to be causally involved in the task (*Esmaeili et al., 2021*), include the primary and secondary whisker sensory cortex (wS1, wS2), the primary and secondary whisker motor cortex (wM1, wM2), the anterior lateral motor cortex (ALM), and the primary tongue-jaw motor cortex (tjM1). We fit the neuronal activity using the same reconstruction method as used for the synthetic dataset. In the model, we simulate the jaw movement of the mouse as a linear readout driven by the model's neural activity. This readout is regressed with the real jaw movement extracted from video footage. The parameter optimization of the behavioral readout is performed jointly with fitting the synaptic weights to the neuronal recordings, see Methods, Optimization and loss function. After training, our reconstructed model can generate neural activity with firing rate distribution, trial-averaged activity, single-trial network

**Table 2.** Trial-matching loss test loss $\mathcal{L}_{\text{trial}}$ of the different reconstruction methods with the real recordings from *Esmaeili et al., 2021* ± indicates the 95% confidence interval.

Unlike in *Table 1*, the same metric cannot be evaluated for the perturbation trials due to the absence of joint recordings and stimulation in this dataset.

| Method name | Real dataset vs reconstructed network |
|---|---|
| bioRNN | 0.76 ± 0.14 |
| σRNN | 0.62 ± 0.12 |
| No sparsity | 0.77 ± 0.15 |
| Non-local inhibition | 0.79 ± 0.15 |
| No Dale's law | 0.68 ± 0.13 |
| No TM | 1.63 ± 0.55 |
| No spike | 0.64 ± 0.13 |

dynamics, and behavioral outcome, which are all consistent with the recordings (see *Figure 3—figure supplement 1*). Before perturbations, we observe again that the σRNN model fits the testing set data better than the bioRNN model (see *Table 2* and *Figure 3—figure supplement 1*).

We then submit the reconstructed σRNNs and bioRNNs models to *perturbation tests*. For the sessions of the in vivo dataset with optogenetic perturbation that we considered, only the behavior of an animal is recorded during inactivation of an area at a given time window (stimulus, delay, or choice periods). For each of the six areas and time windows, we extract the averaged hit frequency

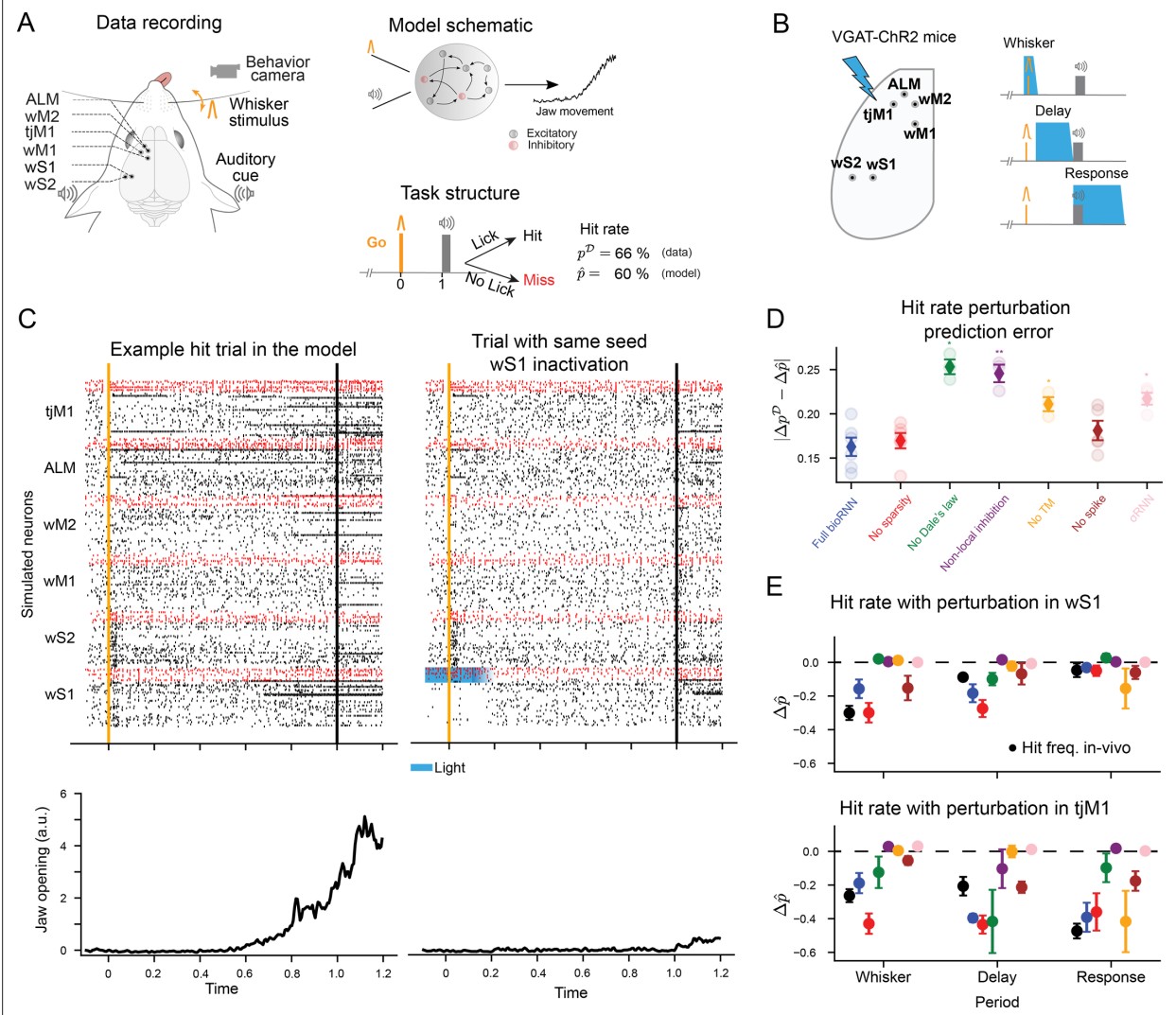

**Figure 3.** Predicting optogenetic perturbations for in vivo electrophysiology data. (**A**) During a delayed whisker detection task, the mouse reports a whisker stimulation by licking to obtain a water reward. Jaw movements are recorded by a camera. Our model simulates the jaw movements and the neural activity from six areas. (**B**) The experimentalists performed optogenetic inactivations of cortical areas (one area at a time) in three temporal windows. (**C**) Example hit trial of a reconstructed network (left). Using the same random seed, the trial turns into a miss trial if we inactivate area wS1 (right, light stimulus indicated by blue shading) during the whisker period by stimulation of inhibitory neurons (red dots). (**D**) Error of the change in lick frequency caused by the perturbation, $\Delta\hat{p}$ is predicted by the model, and $\Delta p^{\mathcal{D}}$ is recorded in mice. Light-shaded circles show individual reconstructed networks with different initializations. The whiskers are the standard error of means. Statistical significance is computed with t-test using the mean change of lick frequency over different network initializations (n=3-6) and is indicated with 0-2 stars corresponding to p-values thresholds: 0.05 and 0.01. (**E**) Examples of $\Delta\hat{p}$ hit rate changes under perturbation for wS1 (Top) and tjM1 (Bottom). The black circles refer to the hit rate change from the recordings, $\Delta p^{\mathcal{D}}$. See *Figure 3—figure supplement 2* for the other areas.

The online version of this article includes the following figure supplement(s) for figure 3:

**Figure supplement 1.** Reconstruction of the real recordings.

**Figure supplement 2.** Hit rate changes under inactivations.

under optogenetic inactivation and attempt to predict this perturbed behavior by inducing the same inactivations to the fitted RNNs. These perturbations are acute spatiotemporal optogenetic inactivations of each area during different time periods (see *Figure 3B*). As an example, we show the effect of an inactivation of wS1 during the whisker period in the model in *Figure 3*. In panel C, we display the simulated trial of a fitted bioRNN with and without perturbations side by side. The two trials are simulated with the same random seed, and this example shows that an early perturbation in wS1 can change a lick decision from hit to miss in the model (*Figure 3C*).

Consistent with the synthetic dataset, we now find with this in vivo dataset that modeling cell-type connectivity yields better prediction of the causal effect of optogenetic perturbation. We denote by $\Delta p^{\mathcal{D}}$ the in vivo change in lick probability across Go trials in response to optogenetic perturbations. The perturbations were performed in different periods for each area in *Esmaeili et al., 2021* (stimulation, delay, or choice periods). For all areas and time windows, we measure the corresponding $\Delta \hat{p}$ in the model. On average, the error change probability obtained with the σRNN model is $|\Delta p^{\mathcal{D}} - \Delta \hat{p}| = 21\%$ which is significantly worse than the bioRNN model's 16% (t-test p-value is 0.014, see *Figure 3D*). As in the synthetic dataset, we find this to be consistent over multiple bioRNN model variants, and we find that imposing Dale's law and local inhibition best explain the improvement in perturbation-robustness. We also measure that the spiking bioRNN predicts the change in lick probability slightly better than the 'No Spike' bioRNN model. Conversely, adding the sparsity prior does not seem to improve the perturbed hit-rate prediction on the real data as seen in the recurrent artificial dataset (RefCirc2) and not in the feedforward case (RefCirc1) as shown in *Figure 2—figure supplement 4*. In this sense, in vivo perturbation testing emerges as a hard test to evaluate modeling strategies combining deep learning and biophysical modeling.

To further analyze the consistency of the perturbations in the model, we can compare the perturbation map showing changes in lick probability obtained from acute inactivation in the data and the model. The *Figure 3—figure supplement 2* summarizes visually which area has a critical role at specific time points. The changes of lick probability in area wS1, ALM, and tjM1 are accurately predicted by the bioRNN. In contrast, our model tends to underestimate the causal effect induced by the inactivations of wS2, wM1, and wM2 (*Figure 3—figure supplement 2*). Overall, our model is consistent with a causal chain of interaction from wS1 to ALM and continuing to tjM1.

## Applications for experimental electrophysiology

With future progress in recording technology and reconstruction methods, network reconstruction may soon predict the effect of optogenetic perturbation with even higher accuracy. In this section, we explore possible consequences and applications for experimental electrophysiology. We demonstrate in the following that (1) perturbation-robust bioRNNs enable us to estimate gradients of the recorded circuits, (2) which in turn enable us to target μ-perturbations in the recorded circuit and optimally increase (or decrease) induced movements in our simulated mouse. The so-called 'recorded circuit' is a bioRNN trained on the in vivo dataset that we use as a proxy experimental preparation. Its mathematical underpinnings enable us to make rigorous theoretical considerations and the design of forward-looking in silico experiments.

### μ-perturbations measure brain gradients

We first prove a mathematical relationship between gradients in the recorded circuit and μ-perturbations. We define the integrated movement as $Y = \sum_t y_t$ where $y_t$ is the movement of the jaw at time $t$ generated by the model, and we denote $A$ as the change of movement caused by the μ-perturbation. If the circuit has well-defined gradients (e.g. say a 'No spike' bioRNN model trained on the in vivo recordings in the previous section), using a Taylor expansion, we find that:

$$\Delta Y = \sum_{i,t \in \mathcal{I}} \frac{dY}{du_i^t} \Delta u_i^t + \epsilon, \tag{1}$$

where $\mathcal{I}$ are the neuron and time indices selected for the optogenetic intervention. The error term $\epsilon$ is negligible when the current $\Delta u_i^t$ induced by the light is small. We first confirm this approximation with numerical visualization in *Figure 4A*: we display movement perturbations $\langle \Delta Y \rangle$ in the circuit with time windows of decreasing sizes ($\langle \cdot \rangle$ indicates a trial average). When the time window is small, and the perturbation is only applied to excitatory or inhibitory cells in *Figure 4A*, one can

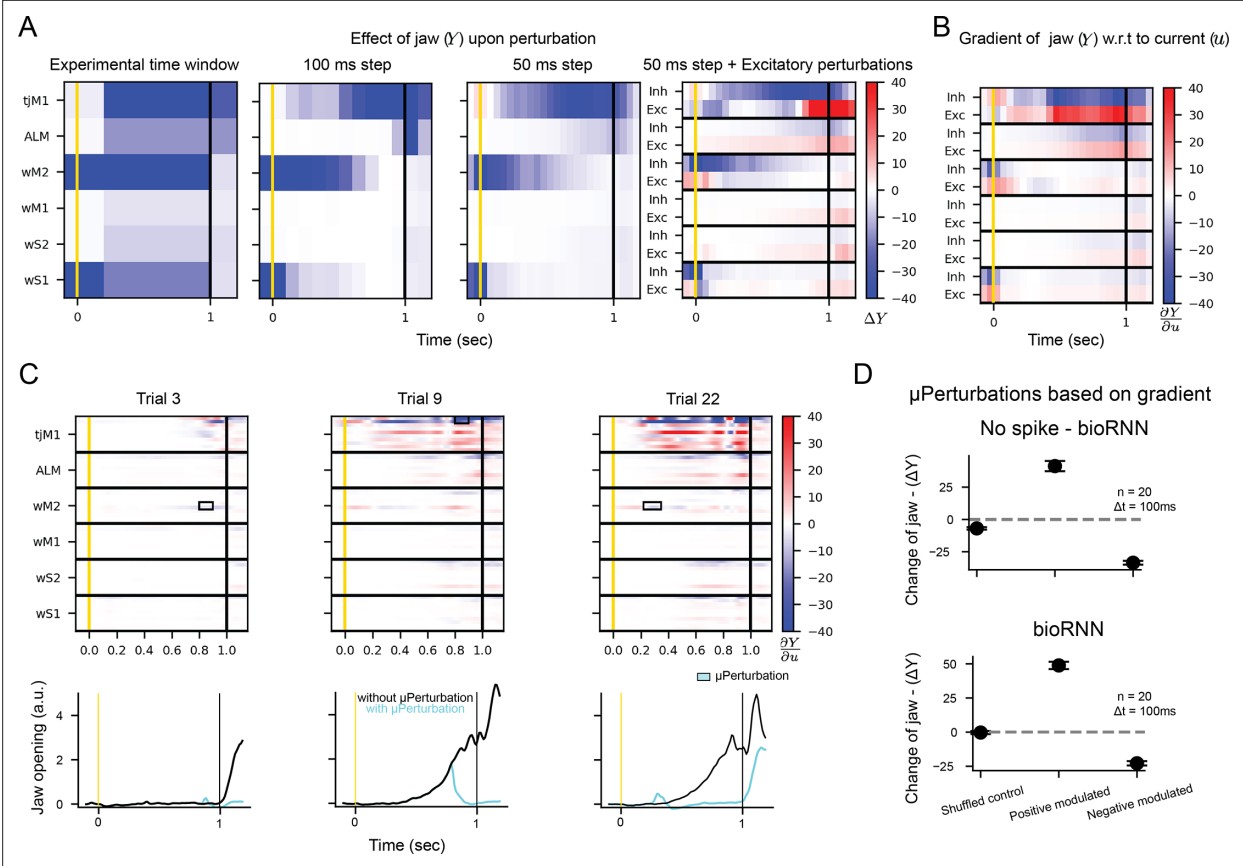

**Figure 4.** Measuring circuit gradients with μ-perturbations. (**A–B**) Numerical verification for *Equation 1*. **A** shows the change of jaw movement $\Delta Y$ following inactivations in a "No Spike" bioRNN. From left to right, we reduce the size of the spatiotemporal window for the optogenetic stimulation. (**B**) Gradient values $\sum_{i,t} \frac{dY}{du}$ that approximate $\Delta Y$ from A using *Equation 1*. (**C–D**) Verification that gradients predict the change of movement on single trials. In **C**, we display the gradients and jaw movement for three different trials, the neurons targeted by the μ-perturbation are boxed and the perturbed jaw movement is blue. Results averaged for every 100ms stimulation windows are shown in (**D**) positive (resp. negative) modulated means that the 20 neurons with highest (resp. lowest) gradients are targeted, random neurons are selected for the shuffled case. The whiskers show the 95% confidence interval.

appreciate visually the similarity with the binned gradient $\langle\sum_{i,t} \frac{dY}{du_i^t}\rangle$ in *Figure 4B*. Proceeding to a quantitative verification of *Equation 1*, we now compare the effect of small perturbations targeting only 20 neurons on a single trial. We use the gradient $\sum_{i,t} \frac{dY}{du_i^t}$ (see *Figure 4C*) to predict the outcome of μ-perturbation as follows: for each trial, and each 100 ms time window, we identify 20 neurons in the model with highest (or lowest) gradients $\sum_{i,t} \frac{dY}{du_i^t}$. We then re-simulate the exact same trial with identical random seed, but induce a μ-perturbation on selected neurons (see rectangles in *Figure 4*). If we target neurons with strongly positive gradients, the perturbed jaw movements are strongly amplified $\Delta Y > 0$; conversely, if we target neurons with negative gradients, the jaw movements are suppressed $\Delta Y < 0$. Although *Equation 1* is only rigorously valid for models with well-defined gradients like the 'No Spike' model, we also confirm in *Figure 4D* that this numerical verification also holds in a spiking circuit model where the gradients are replaced with surrogate gradients (*Neftci et al., 2019*).

An implication of *Equation 1* is that the measurements $\langle\Delta Y\rangle$ that can be recorded in vivo are estimates of the gradients $\langle\sum_{i,t} \frac{dY}{du_i^t}\rangle$ in the recorded circuit. Yet, measuring detailed gradient maps (or perturbation maps) as displayed in *Figure 4* would be costly in vivo as it requires to average $\Delta Y$ over dozens of trials for each spatio-temporal window.

Instead, gradient calculation in a bioRNN model (that was fitted to the experimental preparation) is a rapid mathematical exercise. If the extracted model is valid, then the gradients $\sum_{i,t} \frac{dY}{du_i^t}$ in the

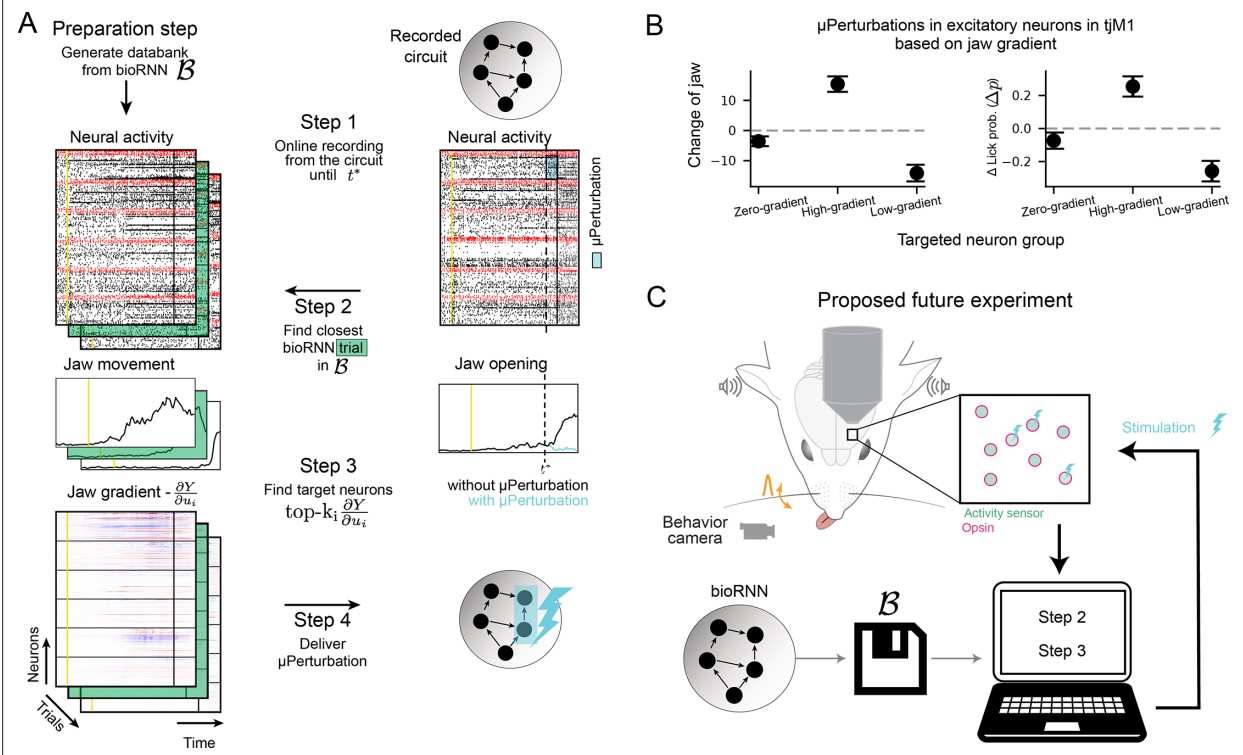

**Figure 5.** Gradient-targeted µ-perturbations could precisely bias an animal behavior. (**A**) Protocol to deliver an optimal µ-perturbation on the experimental preparation based on jaw gradients. (Step 1) The circuit is recorded until stimulation time $t^*$. (Step 2) The closest bioRNN trial to the ongoing recorded trial is retrieved from the databank $\mathcal{B}$. (Step 3) We select the neurons with the highest (or lowest) gradient value for the µ-perturbation. (Step 4) The µ-perturbation is delivered at $t^*$. (**B**) Effect of the µ-perturbation using the artificial setup (**A**) under different light protocols. Practically, for 'High gradient', we keep step 3 as it is, for 'Low gradient', we change the sign of the gradient, and for 'Zero gradient', we pick the 40 neurons with lowest gradient norm. The whisker indicate the 95% confidence interval. (**C**) Speculative schematic of a close-up setup implementing the protocol (**A**) inspired by the all optical 'read-write' setup from *Aravanis et al., 2007*; *Packer et al., 2015*.

bioRNN approximate (1) the effect of µ-perturbations $\Delta Y$ in the experimental preparation; (2) the gradient $\sum_{i,t} \frac{dY}{du_i^t}$ in the recorded circuit.

## Targeting in vivo µ-perturbations with bioRNN gradients

Building on this theoretical finding, we build a speculative experimental setup where the bioRNN gradients are used to target a µ-perturbation and increase (or decrease) the movements $Y$ in the experimental preparation in real time. We show a schematic of this speculative closed-loop experiment in *Figure 5C* extending contemporary read-write electrophysiology setups (*Packer et al., 2015*; *Adesnik and Abdeladim, 2021*; *Grosenick et al., 2015*; *Papagiakoumou et al., 2020*). We demonstrate in silico in *Figure 5A–B* how this experiment could use bioRNN gradients to bias the simulated mouse movement $Y$. As a preparation step, and before applying perturbations, we assume that the bioRNN is well fitted to the recorded circuit and we collect a large databank $\mathcal{B}$ of simulated trials from the fitted bioRNN. Then in real-time, we record the activity from the experimental preparation until the time $t^*$ at which the stimulation will be delivered (Step 1 in *Figure 5A*, $t^*$ is 100 ms before the decision period). Rapidly, we find the trial with the closest spike trains in the databank of simulated trials (Step 2) and use the corresponding gradient maps to target neurons with the highest gradient $\frac{dY}{du}$ in the model (Step 3). The targeted stimulation is then delivered immediately at $t^*$ to the experimental preparation (Step 4). When testing this in silico on our artificial experimental preparation, we show in *Figure 5C* that this approach can bias the quantity of jaw movement $Y$ driven by the circuit in

a predictable way. The amount of movement is consistently augmented if we target neurons with the highest $\frac{dY}{du}$ (or diminished if we target neurons with the lowest $\frac{dY}{du}$).

## Discussion

Finding the right level of detail to model recorded phenomena has sparked intensive debates in computational neuroscience. When the goal is to achieve the strongest predictive power, generalist deep learning models have proven successful across many scientific disciplines, questioning how biophysical modeling plays a role in this context. Our results show that *perturbation testing* is a new approach to evaluate the implementation of biophysical features in a deep learning system. Our key finding about *perturbation testing* relies on the difficulty for deep learning models to predict the effect of optogenetic perturbations out-of-distribution (meaning, the perturbed trials are not available in the training set of the data-constrained model). We see that standard deep learning RNNs generalize poorly to perturbed trials, even when they achieved the best fit on the unperturbed test set. In contrast, this is alleviated with our bioRNN, which implements biophysical constraints that are relevant to the nature of the perturbation. In our case, modeling cell type connectivity is crucial because the optogenetic perturbations are targeted to these genetically encoded cell types. In this sense, we believe that these features were successful on the perturbation tests because they are central to modeling the perturbation of the deep learning system. Perturbation testing emerges as a quantitative tool to search for data-constrained models beyond two standard types of incomplete brain models in computational neuroscience: (1) physiologically detailed models intended to explain brain mechanisms but do not enable powerful quantitative predictions; (2) deep learning models with high predictive power but capturing a wrong biophysical mechanism, causing erroneous generalizations. We view our work as a simple and reasonable way to combine deep learning and biophysical modeling, while rigorously evaluating the combined models.

Our reconstruction method and modeling choices when building the data-constrained bioRNN are innovative and are validated on *perturbation tests*. We achieve a partial reconstruction of the sensory-motor pathway in the mouse cortex during a sensory detection task from electrophysiology data. The model is optimized to explain electrophysiological recordings and generalizes better than standard models to in vivo optogenetic interventions. We found unambiguously that anatomically informed sign and connectivity constraints for dominant excitatory and inhibitory cell types improve the model robustness to optogenetic perturbations. We also find that assuming that inhibitory connections are short and do not project to other areas is crucial to pass our optogenetic *Perturbation test*. Modeling spiking neuron dynamics and adding a sparsity prior yielded more nuanced results and was not decisive, showing that making a difference on *Perturbation testing* is challenging. In hindsight, we conclude that adding biological constraints becomes beneficial when (1) they model the interaction between the circuit and the perturbation mechanism; (2) their implementation should not impair the efficiency of the optimization process.

Broadly speaking, this hindsight is also supported by other results elsewhere in neuroscience. For instance, biologically inspired topological networks having higher correlation for neighboring neurons are more consistent with comparable causal interventions in the Monkey's visual system (*Schrimpf et al., 2024*), and detailed cell-type distribution and connectome improve models of vision in the fly brain (*Lappalainen et al., 2023*; *Cowley et al., 2024*). For future work, there is a dense knowledge of unexploited physiological data at the connectivity, laminar, or cell-type level that could be added to improve a cortical model like ours (*Harris et al., 2019*; *Liu et al., 2022*; *Udvary et al., 2022*; *Staiger and Petersen, 2021*; *Rimehaug et al., 2023*). By submitting the extended models to the relevant *perturbation tests*, it becomes possible to measure quantitatively the goodness of their biological mechanism implementations. We do not rule out that significant improvements on *perturbation tests* can also be achieved with other means (e.g. by training deep learning architectures *Azabou et al., 2024*; *Pandarinath et al., 2018*; *Ye et al., 2023* on larger datasets to enable generalization, or with generic regularization techniques like low-rank connectivity *Dubreuil et al., 2022*; *Valente et al., 2022*). However, in a similar way as the σRNN was a priori an excellent predictor on our initial test set, any powerful brain model will likely have failure modes that can be well characterized and measured with an appropriate perturbation test. So *perturbation tests* could become a central component of an iterative loop to identify needed data collection or model improvements towards robust brain models.

To highlight the importance of perturbation-robust circuit models, we have discussed possible implications for experimental neuroscience in section 2.3. We build the RNN twin of the biological circuit from unperturbed electrode recordings. By implementing the correct biophysical constraints, the RNN becomes perturbation robust (i.e. it predicts the effect of causal perturbation) even without including perturbation data in the RNN training. We then demonstrated in silico that gradients of this RNN produce sensitivity maps to target micro-stimulation of the biological circuit. As a result, we could design a hypothetical closed-loop setup combining read-write electrophysiology with a brain model to influence the brain activity or behavior, having potentially important practical and ethical consequences. More conceptually, we have shown theoretically that the gradients of a perturbation-robust RNN are also consistent with the gradients of the recorded biological circuits. In perspective with the foundational role of gradients in machine learning theory (*LeCun et al., 2015*; *Richards and Kording, 2023*), it enables the measurement of 'brain gradients' and lays a computational link between in vivo experimental research and decades of theoretical results on artificial learning and cognition.

## Methods

## Mathematical toy model of the difficult causal inference between H1 and H2

Let's consider two simplistic mathematical models that both depend on two binary random variables $A$ and $B$ which represent that putative area A is active as $A = 1$ and area B as $B = 1$. With this notation, we can construct two hypothetical causal mechanisms $H1$ ('a feedforward hypothesis') and $H2$ ('a recurrent hypothesis'), which are radically different. The empirical frequency $p(A, B)$ of the outcome does not allow us to differentiate whether the system was generated by a feedforward mechanism $H1$ or a recurrent mechanism $H2$. Schematically, we can represent the two mechanism hypotheses as follows:

$$(H1) \quad A \longrightarrow B, \tag{2}$$

$$(H2) \quad A \longleftrightarrow B. \tag{3}$$

For hypothesis $H1$: we assume that external inputs are driving the activity of area A such that $A = 1$ is active with probability $p_0$, and there are strong feed-forward connections from $A$ to $B$ causing systemically $B = 1$ as soon as $A = 1$. Alternatively, in $H2$, we assume that areas A and B receive independent external inputs with probability $p_1 = 1 - \sqrt{1 - p_0}$. Each of these two inputs is sufficient to cause $A = 1$ or $B = 1$, and the two areas are also strongly connected, so $A = 1$ always causes $B = 1$ and vice versa. Under these hypothetical mechanisms $H1$ and $H2$, one finds that the empirical probability table $p(A, B)$ is identical as seen in the proof at the end of subsection: $p_{H2}(A = 1, B = 1) = 2p_1 - p_1^2 = p_0$ ('Hit trial', both areas are active), $p(A = 0, B = 0) = 1 - p_0$ ('Miss trial', the areas are quiescent). In both cases, the possibility that only one area is active is excluded by construction. So for any $A$ and $B$ $p_{H1}(A, B) = p_{H2}(A, B)$ and in other words, even if we observe an infinite number of trials and compute any statistics of the binary activations $A$ and $B$, discriminating the two possible causal interactions (H1 versus H2) is impossible.

A solution to discriminate between hypotheses $H1$ and $H2$ is to induce a causal perturbation. We can discriminate between our two hypotheses if we can impose a perturbation that forces the inactivation of area $B$ in both mathematical models. In mathematical terms, we refer to the *do* operator from causality theory. Under the feedforward mechanism $H1$ and inactivation of $B$, $A$ is not affected $p_{H1}(A = 1 | do(B = 0)) = p_0$. Under the recurrent hypothesis, $H2$, and inactivation of $B$, $A$ is activated only by its external input such that $p_{H2}(A = 1 | do(B = 0)) = p1 \neq p_0$. So the measurement of the frequency of activation of area $A$ under inactivation of $B$ can discriminate between $H1$ and $H2$ which illustrates mathematically how a causal perturbation can be decisive to discriminate between those two hypothetical mechanisms.

Proof: Let $a$ and $b$ denote the binary exteral inputs into A and B, we have: $p_{H2}(A = 1, B = 1) = \sum_{a,b} p(A = 1, B = 1 | a, b) p(a, b) = p(a = 1, b = 1) + p(b = 0, a = 1) + p(b = 1, a = 0)$ then if we have that $p(A = 1, B = 1 | a, b)$ is 0 or 1, and $p(a = 1) = (b = 1) = p1$. With the independence between $a$ and $b$ we find: $p(A = 1, B = 1) = 2p_1 - p_1^2 = p_0$.

## Neuron and jaw movement model

We model neurons as leaky-integrate and fire (LIF) neurons. The output of every neuron $j$ at time $t$ is a binary outcome $z_j^t$ (spike if $z_j^t = 1$, no spike if $z_j^t = 0$) generated from its membrane voltage $v_j^t$. The following equations give the dynamics of the membrane voltage $v_j^t$:

$$v_j^t = \alpha_j v_j^{t-1} + (1 - \alpha_j)u_j^t - v_{\text{thr},j}z_j^{t-1} + \xi_j^t \tag{4}$$

$$u_j^t = \sum_{d,i} W_{ij}^{rec,d}\frac{z_i^{t-d}}{\delta t} + \sum_i W_{ij}^{in}\frac{x_i^t}{\delta t} \tag{5}$$

where $W_{ij}^d$, and $W_{ij}^{in,d}$ are the recurrent and input weight matrices. The timestep of the simulation $\delta t$ is 2ms when we simulate the real dataset and 1ms otherwise. The superscript $d$ denotes the synaptic delay; every synapse has one synaptic delay of either 2 or 3ms. With $\alpha_j = \exp\left(-\frac{\delta t}{\tau_{m,j}}\right)$, we define the integration speed of the membrane voltage, where $\tau_m = 30$ ms for excitatory and $\tau_m = 10$ ms for inhibitory neurons. The noise source $\xi_j^t$ is a Gaussian random variable with zero mean and standard deviation $\beta_j v_{\text{thr},j}\sqrt{\delta t}$ ($\beta_j$ is typically initialized at 0.14). The input $x_i^t$ is a binary pulse signal with a duration of 10ms. For the real dataset, we have two binary pulse input signals, one for the whisker deflection and one for the auditory cue. The spikes are sampled with a Bernoulli distribution $z_j^t \sim \mathcal{B}(\exp(\frac{v_j^t - v_{\text{thr},j}}{v_0}))$, where $v_0$ is the temperature of the exponential function and $v_{thr,j}$ is the effective membrane threshold. After each spike, the neuron receives a reset current with an amplitude of $v_{trh,j}$ and enters an absolute refractory period of 4ms, during which it cannot fire.

For networks fitted to the real dataset, we also simulate the jaw movement. The jaw movement trace $y$ is controlled by a linear readout from the spiking activity of all excitatory neurons. Specifically, $y$ is computed as $y = \exp(\tilde{y}) + b$, where $b$ is a scaling parameter and $\tilde{y}^t$ is given by $\tilde{y}^t = \alpha_{jaw}\tilde{y}^{t-1} + (1 - \alpha_{jaw})\sum_{d,j} W_j^{jaw}z_j^{t-d}$. Here, $W_j^{jaw}$ is the output weight matrix (linear readout) for the jaw, and $\tau_{jaw} = 5$ ms defines $\alpha_{jaw} = \exp(-\frac{\delta t}{\tau_{jaw}})$, which controls the integration velocity of the jaw trace.

## Session-stitching and network structure

As in *Sourmpis et al., 2023*, we simulate multi-area cortical neuronal activity fitted to electrophysiology neural recordings. Before we start the optimization, we define and fix each neuron's area and cell type in the model by uniquely assigning them to a neuron from the recordings. For the real dataset from *Esmaeili et al., 2021*, the cell type is inferred from the cell's action potential waveform (with fast-spiking neurons classified as inhibitory and regular-spiking neurons as excitatory). Most electrophysiology datasets include recordings from multiple sessions, and our method would typically require simultaneous recordings of all neurons. To address this challenge, similarly to *Sourmpis et al., 2023* we use the technique called 'session-stitching' which allows neighboring modeled neurons to be mapped with neurons recorded across multiple sessions. This effectively creates a 'collage' that integrates data from multiple sessions within our model. This approach has practical implications for our optimization process. Specifically, the trial-matching loss includes a term for each session, with the overall loss calculated as the average across all sessions (see, Optimization and loss function).

For both the real and the synthetic datasets, we simulate each area with 250 LIF neurons and impose that each area has 200 excitatory neurons and 50 inhibitory. Respecting the observation that inhibitory neurons mostly project in the area that they belong to *Tamamaki and Tomioka, 2010*; *Markram et al., 2004*, we don't allow for across-area inhibitory connections. The 'thalamic' input is available to every neuron of the circuit, and the 'motor' output for the real dataset, that is jaw movement, is extracted with a trained linear readout from all the excitatory neurons of the network, see (Neuron and jaw movement model).

## Reference circuits for hypotheses 1 and 2

To build a synthetic dataset that illustrates the difficulty of separating the feedforward (H1) and recurrent hypotheses (H2), we construct two reference spiking circuit models RefCirc1 and RefCirc2. The two networks consist of two areas A and B, and their activity follows the hard causal inference problem from method (Mathematical toy model of the difficult causal inference between H1 and H2), making it hard to distinguish A1 and A2 when recording the co-activation of A and B. Moreover, to make the

problem even harder, the two networks are constructed to make it almost impossible to distinguish between H1 and H2 with dense recordings: the two circuits are designed to have the same PSTH and single-trial network dynamics despite their structural difference; one is feedforward and the other is recurrent.

To do so, RefCirc1 and 2 are circuit models that start from random network initializations following the specifications described in Methods sections Neuron and jaw movement model and Session-stitching and network structure. The only difference is that we do not allow feedback connections from A to B in RefCirc1; the construction below is otherwise identical. The synaptic weights of the two circuits are optimized with the losses described in Methods (Optimization and loss function) to fit the identical target statistics in all areas: the same PSTH activity for each neuron and the same distribution of single-trial network dynamics. The target statistics are chosen so the activity in RefCirc1 and 2 resembles kinematics and statistics from a primary and a secondary sensory area. The baseline firing rates of the neurons are dictated by the target PSTH distribution and it follows a log-normal distribution, with excitatory neurons having a mean of 2.9 Hz and a standard deviation of 1.25 Hz and inhibitory neurons having a mean of 4.47 Hz and a standard deviation of 1.31 Hz. The distribution of single-trial activity is given by the targeted single-trial dynamics: in RefCirc1 and 2, the areas A and B respond to input 50% of the time with a transient population average response following a double exponential kernel characterized by $\tau_{rise}$ = 5ms and $\tau_{fall}$ = 20ms. Mimicking a short signal propagation between areas, these transients have a 4ms delay in area A and 12ms delay in B (relative to the onset time of the stimulus). To impose a 'behavioral' hit versus miss distribution that could emerge from a feedforward and recurrent hypothesis (see Methods, Mathematical toy model of the difficult causal inference between H1 and H2), the targeted population-averaged response of each trial is either a double-exponential transient in both area A and B ('Hit trials'), or remains at a baseline level in both areas ('Miss trials') in the remaining trials. At the end of the training, we verified that RefCirc1 and RefCirc2 generate very similar network activity in the absence of perturbation (see *Figure 1—figure supplement 1*). The circuits are then frozen and used to generate the synthetic dataset. We generate 2000 trials from these RefCircs, 1000 of which are used for the training set and 1000 for the testing set.

## Optimization and loss function

The optimization method we use to fit our models is back-propagation through time (BPTT). To overcome the non-differentiability of the spiking function, we use surrogate gradients (*Neftci et al., 2019*). In particular, we use the piece-wise linear surrogate derivative from *Bellec et al., 2018b*. For the derivative calculations, we use $\frac{v_j^t - v_{\text{thr},j}}{v_0}$ and not $\exp(\frac{v_j^t - v_{\text{thr},j}}{v_0})$. We use sample-and-measure loss functions that rely on summary statistics, as in *Bellec et al., 2021*; *Sourmpis et al., 2023*, to fit the networks to the data. Our loss function has two main terms: one to fit the trial-averaged activity of every neuron ($\mathcal{L}_{\text{neuron}}$), and one to fit the single trial population average activity ($\mathcal{L}_{\text{trial}}$), $\mathcal{L} = \mathcal{L}_{\text{neuron}} + \mathcal{L}_{\text{trial}}$. The two terms of the loss function are reweighted with a parameter-free multi-task method (*Défossez et al., 2023*) that enables the gradients to have comparable scales.

As in *Sourmpis et al., 2023*: (1) To calculate the trial-averaged loss, we first filter the trial-averaged spiking activity $\mathcal{T}_{\text{neuron},j}^t(z) = \frac{1}{K} \sum_k z_{j,k}^t * f$ using a rolling average window ($f$) of 8 ms. We then normalize it by the trial-averaged filtered data activity, ($z^{\mathcal{D}}$ are recorded spike trains)

$$\mathcal{T}_{\text{neuron},j}^{\prime t}(z) = (\mathcal{T}_{\text{neuron},j}^t(z) - \langle \mathcal{T}_{\text{neuron},j}^t(z^{\mathcal{D}}) \rangle_t)/(\sigma_t(\mathcal{T}_{\text{neuron},j}^t(z^{\mathcal{D}}))), \tag{6}$$

where $\langle . \rangle_t$ is the time average, and $\sigma_t$ the standard deviation over time. The trial-averaged loss function is defined as:

$$\mathcal{L}_{\text{neuron}} = \sum_j^N \sum_t^T \| \mathcal{T}_{\text{neuron},j}^{\prime t}(z) - \mathcal{T}_{\text{neuron},j}^{\prime t}(z^{\mathcal{D}}) \|^2 , \tag{7}$$

where $T$ is the number of time points in a trial and $N$ is the number of neurons. For the real dataset, where we want to fit also the jaw movement, we have an additional term for the trial-averaged filtered and normalized jaw, $\| \sum_t^T \mathcal{T}_{\text{neuron}}^{\prime t}(y) - \mathcal{T}_{\text{neuron}}^{\prime t}(y^{\mathcal{D}}) \|^2$, where $y$ is the simulated jaw movement and $y^{\mathcal{D}}$ the recorded jaw movement.

To calculate the trial-matching loss, we first filter the population-average activity of each area $A$, $\mathcal{T}_{A,k}^t(z) = \frac{1}{|A|} \sum_{j \in A} z_{j,k}^t * f$, using a rolling average window of 32ms. We then normalize

it by the population-averaged filtered activity of the same area from the recordings, $\mathcal{T}_{A,k}^{\prime t}(z) = (\mathcal{T}_{A,k}^{t}(z) - \langle\mathcal{T}_{A,k}^{t}(z^{\mathcal{D}})\rangle_k)/\sigma_k(\mathcal{T}_{A,k}^{t}(z^{\mathcal{D}}))$, and concatenate all the areas that were simultaneously recorded, $\mathcal{T}_{\text{trial},k}^{\prime t}(z) = (\mathcal{T}_{A1,k}^{\prime t}, \mathcal{T}_{A2,k}^{\prime t})$, where $\langle.\rangle_k$ is the trial average, and $\sigma_k$ the standard deviation over trials. The trial-matching loss is defined as:

$$\mathcal{L}_{\text{trial}} = \min_{\pi} \sum_k^K \sum_t^T \|\mathcal{T}_{\text{trial},k}^{\prime t}(z) - \mathcal{T}_{\text{trial},\pi(k)}^{\prime t}(z^{\mathcal{D}})\|^2 , \tag{8}$$

where $\pi$ is an assignment between pairs of $K$ recorded and generated trials $\pi : \{1, \ldots K\} \rightarrow \{1, \ldots K\}$. Note that the minimum over $\pi$ is a combinatorial optimization that needs to be calculated for every evaluation of the loss function. For the real dataset, we consider the jaw movement as an additional area, and we concatenate it to the $\mathcal{T}_{\text{trial},k}^{\prime t} = (\mathcal{T}_{A1,k}^{\prime t}, \mathcal{T}_{A2,k}^{\prime t}, \mathcal{T}_{jaw,k}^{\prime t})$.

Based on this loss function, we optimize the following parameters: $W_{ij}^{rec,d}$, $W_{ij}^{in,d}$, $v_{thr,j}$, and $\beta$ for the RefCircs. For the RNNs, we optimize only the recurrent connectivity $W_{ij}^{rec,d}$, and the rest are fixed from the RefCircs. For the real dataset, in addition to the parameters optimized in the RefCircs, we also optimize the jaw's linear readout $W_j^{jaw}$ and its scaling parameter $b$.

## Implementing Dale's law and local inhibition

In our network, the recurrent weights $W^{rec}$ are computed as the elementwise product of two matrices: $\tilde{W}^{rec}$, which encodes the strength of synaptic efficacies and is always positive, and $W_{sign}^{rec}$, which has a fixed sign determined by the neurotransmitter type of the presynaptic neuron and $|W_{sign}^{rec}| = 1$:

$$W^{rec} = \tilde{W}^{rec} \circ W_{sign}^{rec} \tag{9}$$

To enforce Dale's law during optimization, we set any negative values of $\tilde{W}^{rec}$ to zero at each iteration as in *Bellec et al., 2018a*. Similarly, to constrain only local inhibitory connections during optimization, we zero out any changes in the synaptic efficacies of across-areas inhibitory connections at each iteration. In simplified models, Dale's law or the local inhibition constraint can be disrupted by omitting this correction step.

The success of the network optimization highly depends on the initialization of the recurrent weight matrices. To initialize signed matrices, we follow the theoretical (*Rajan and Abbott, 2006*) and practical insights (*Bellec et al., 2018b*; *Cornford et al., 2020*) developed previously. After defining the constraints on the weight signs $W_{sign}^{rec}$, the initialization amplitude $\tilde{W}^{rec}$ for each target neuron is adjusted to a zero-summed input weights (the sum of incoming excitatory inputs is equal to the sum of inhibitory inputs). Then the weight amplitude is re-normalized by the modulus of its largest eigenvalue of $W^{rec}$, so all the eigenvalues of this matrix $W^{rec}$ have modulus 1 or smaller.

## Stopping criterion for the optimization

For the synthetic dataset, we train the models for 4000 gradient descent steps. For the real dataset, due to limited data and a noisy test set, we select the final model based on the optimization step that yields the best trial-type accuracy (closest to the trial-type accuracy from the data), derived from the jaw trace and whisker stimulus, along with the highest trial-matched Pearson correlation between the model and the recordings.

## Sparsity regularization

There is a plethora of ways to enforce sparsity. In this work, we use weight regularization. In particular, we use the p- norm with p=1/2 of the recurrent and input weights that promote a high level of sparsity (*Xu et al., 2012*). To avoid numerical instabilities, we apply this regularization only for synaptic weights above $\alpha$ and prune all synapses below $\alpha$. (we set $\alpha = 1e^{-7}$). The regularized loss function becomes:

$$\mathcal{L}_{all} = \mathcal{L} + \lambda_1 \left\| W^{in} \right\|_{\frac{1}{2}}^{\frac{1}{2}} + \lambda_2 \left\| W^{rec,d} \right\|_{\frac{1}{2}}^{\frac{1}{2}} + \lambda_3 \left\| W^{\text{across},d} \right\|_{\frac{1}{2}}^{\frac{1}{2}} , \tag{10}$$

where $W^{\text{across},d}$ are the connections from one area to the other.

For the synthetic dataset, we choose the level of across-area sparsity by performing a small grid search for $\lambda_3$. In particular, the sparsity level $\lambda_3$ is the maximum value $\lambda_3$ where the performance remains as good as without sparsity, see *Figure 2—figure supplement 2*. For the real dataset, we use the same value $\lambda_3$ as the one we found for the full reconstruction method of bioRNN1.

## Perturbation test of in silico optogenetics

In systems neuroscience, a method to test causal interactions between brain regions uses spatially and temporally precise optogenetic activations or inactivations (*Esmaeili et al., 2021*; *Guo et al., 2014*). Usually, inactivations refer to the strong activation of inhibitory neurons for cortical areas. These inhibitory neurons have strong intra-area connections that effectively 'silence' their local neighborhood (*Helmstaedter et al., 2009*).

Our model can simulate these perturbations and allow us to compare the causal mechanisms of two networks based on their responses to optogenetic perturbations. We implement activations and inactivations as a strong input current to all the neurons in one area's excitatory or inhibitory population. For the RefCircs and reconstructed RNNs, we use a transient current that lasts 40ms, from 20ms before to 20ms after the input stimulus. The strength of the current (light power) varies until there is an effect in the full reconstruction method bioRNN1. For the synthetic dataset in *Figure 2* (except for panel D), we inject a current of $\Delta u_i^t = 0.08$ into excitatory neurons for activations and $\Delta u_i^t = 1$ into inhibitory neurons for inactivations. For the real dataset, we perform optogenetics inactivations in three different periods. As in *Esmaeili et al., 2021*, we silence the cortical circuit during the whisker presentation, the time between the whisker and auditory stimulus, or when the animal was licking for the reward. In particular, we use transient currents to the inhibitory neurons during (i.) 100 ms before and after the whisker presentation, (ii.) 100 ms after the whisker presentation till 100ms before the onset of the auditory cue, and (iii.) after the auditory cue till the end of our simulation. For cases (i.) and (ii.), we linearly decreased the strength of the current to avoid rebound excitation. The light power is chosen so that our model has the best results in reproducing the lick probability of the recordings. It is important to mention that the perturbation data are not used to optimize the network but to test whether the resulting network has the same causal interactions with the recordings.

For the RefCircs and bioRNNs, we evaluate the effect of the perturbations directly from the neural activity. We use the distance of network dynamics $\mathcal{L}_{\text{trial}}$ to compare the two perturbed networks. For the real dataset, we compare the effect of the inactivations on the behavior; as behavior here, we mean whether the mouse/model licked. We classify the licking action using a multilayer perceptron with two hidden layers with 128 neurons each. The classifier is trained with the jaw movement of the real dataset, which was extracted from video filming using custom software, to predict the lick action, which was extracted from a piezo sensor placed in the spout. This classifier predicted lick correctly 94% of the time. We then used the same classifier on the jaw movement from the model to determine whether there was a 'lick' or not. For the comparisons in both the artificial and real datasets, we trained multiple models with different random seeds for each variant and aggregated the results. The different random seeds affect both the weight initialization and the noise of our model. In particular, we used from three to six different random seeds for each different model variant.

## Acknowledgements

We thank Alireza Modirshanechi, Shuqi Wang, and Tâm Nguyen for their valuable feedback on the manuscript. We are grateful to Vahid Esmaeili for collecting the dataset and ongoing support throughout this project. This research is supported by the Sinergia project CRSII5_198612, the Swiss National Science Foundation (SNSF) project 200020_207426 awarded to WG, SNSF projects TMAG-3_209271 and 31003 A_182010 awarded to CP, and the Vienna Science and Technology Fund (WWTF) project VRG24-018 awarded to GB.

## Additional information

### Funding

| Funder | Grant reference number | Author |
|---|---|---|
| Schweizerischer Nationalfonds zur Förderung der Wissenschaftlichen Forschung | CR-SII5_198612 | Wulfram Gerstner |
| Schweizerischer Nationalfonds zur Förderung der Wissenschaftlichen Forschung | 00020_207426 | Wulfram Gerstner |
| Schweizerischer Nationalfonds zur Förderung der Wissenschaftlichen Forschung | TMAG-3_209271 | Carl CH Petersen |
| Schweizerischer Nationalfonds zur Förderung der Wissenschaftlichen Forschung | 31003A_182010 | Carl CH Petersen |
| Vienna Science and Technology Fund | VRG24-018 | Guillaume Bellec |

The funders had no role in study design, data collection and interpretation, or the decision to submit the work for publication.

### Author contributions

Christos Sourmpis, Conceptualization, Data curation, Software, Validation, Investigation, Visualization, Methodology, Writing – original draft, Writing – review and editing; Carl CH Petersen, Conceptualization, Data curation, Supervision, Funding acquisition, Writing – review and editing; Wulfram Gerstner, Conceptualization, Supervision, Funding acquisition, Writing – review and editing; Guillaume Bellec, Conceptualization, Data curation, Software, Formal analysis, Supervision, Funding acquisition, Validation, Investigation, Visualization, Writing – original draft, Writing – review and editing

### Author ORCIDs

Christos Sourmpis ⓘ https://orcid.org/0009-0007-0519-1116
Carl CH Petersen ⓘ https://orcid.org/0000-0003-3344-4495
Guillaume Bellec ⓘ https://orcid.org/0000-0001-7568-4994

Reviewer #1 (Public review): https://doi.org/10.7554/eLife.106827.3.sa1
Reviewer #2 (Public review): https://doi.org/10.7554/eLife.106827.3.sa2
Author response https://doi.org/10.7554/eLife.106827.3.sa3

## Additional files

### Supplementary files

MDAR checklist

### Data availability

The code for this project is open-sourced and published at https://github.com/Sourmpis/Biologically-Informed (copy archived at *Sourmpis, 2026*). The dataset for the artificial dataset can be downloaded/generated on our code repository. The in vivo dataset was published openly for the previous publication *Esmaeili et al., 2021*. The dataset is accessible at: https://doi.org/10.5281/zenodo.4720013.

The following previously published dataset was used:

| Author(s) | Year | Dataset title | Dataset URL | Database and Identifier |
|---|---|---|---|---|
| Esmaeili V | 2021 | Data set for "Rapid suppression and sustained activation of distinct cortical regions for a delayed sensory-triggered motor response" | https://doi.org/10.5281/zenodo.4720013 | Zenodo, 10.5281/zenodo.4720013 |

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
