## [Editor Report · eLife Assessment]

This **important** study demonstrates the significance of incorporating biological constraints in training neural networks to develop models that make accurate predictions under novel conditions. By comparing standard sigmoid recurrent neural networks (RNNs) with biologically constrained RNNs, the manuscript offers **compelling** evidence that biologically grounded inductive biases enhance generalization to perturbed conditions. This manuscript will appeal to a wide audience in systems and computational neuroscience.

---

## [Referee Report · Reviewer #1 (Public review)]

This manuscript introduces a biologically informed RNN (bioRNN) that predicts the effects of optogenetic perturbations in both synthetic and in vivo datasets. By comparing standard sigmoid RNNs (σRNNs) and bioRNNs, the authors make a compelling case that biologically grounded inductive biases improve generalization to perturbed conditions. This work is innovative, technically strong, and grounded in relevant neuroscience, particularly the pressing need for data-constrained models that generalize causally.

Comments on revisions:

The authors have addressed all my concerns.

---

## [Referee Report · Reviewer #2 (Public review)]

Sourmpis et al. present a study in which the importance of including certain inductive biases in the fitting of recurrent networks is evaluated with respect to the generalization ability of the networks when exposed to untrained perturbations.

The work proceeds in three stages:

(i) a simple illustration of the problem is made. Two reference (ground-truth) networks with qualitatively different connectivity, but similar observable network dynamics, are constructed, and recurrent networks with varying aspects of design similarity to the reference networks are trained to reproduce the reference dynamics. The activity of these trained networks during untrained perturbations is then compared to the activity of the perturbed reference networks. It is shown that, of the design characteristics that were varied, the enforced sign (Dale's law) and locality (spatial extent) of efference were especially important.

(ii) The intuition from the constructed example is then extended to networks that have been trained to reproduce certain aspects of multi-region neural activity recorded from mice during a detection task with a working-memory component. A similar pattern is demonstrated, in which enforcing the sign and locality of efference in the fitted networks has an influence on the ability of the trained networks to predict aspects of neural activity during unseen (untrained) perturbations.

(iii) The authors then illustrate the relationship between the gradient of the motor readout of trained networks with respect to the net inputs to the network units, and the sensitivity of the motor readout to small perturbations of the input currents to the units, which (in vivo) could be controlled optogenetically. The paper is concluded with a proposed use for trained networks, in which the models could be analyzed to determine the most sensitive directions of the network and, during online monitoring, inform a targeted optogenetic perturbation to bias behavior.

The authors do not overstate their claims, and in general, I find that I agree with their conclusions.

---

## [Author Response]

The following is the authors’ response to the original reviews.

**Public Reviews:**

**Reviewer #1 (Public review)**
Major:(1) In line 76, the authors make a very powerful statement: 'σRNN simulation achieves higher similarity with unseen recorded trials before perturbation, but lower than the bioRNN on perturbed trials.' I couldn't find a figure showing this. This might be buried somewhere and, in my opinion, deserves some spotlight - maybe a figure or even inclusion in the abstract.

We agree with the reviewer that these results are important. The failure of σRNN on perturbed data could be inferred from the former Figures 1E, 2C-E, and 3D. Following the reviewers' comments, we have tried to make this the most prominent message of Figure 1, in particular with the addition of the new panel E. We also moved Table 1 from the Supplementary to the main text to highlight this quantitatively.

(2) It's mentioned in the introduction (line 84) and elsewhere (e.g., line 259) that spiking has some advantage, but I don't see any figure supporting this claim. In fact, spiking seems not to matter (Figure 2C, E). Please clarify how spiking improves performance, and if it does not, acknowledge that. Relatedly, in line 246, the authors state that 'spiking is a better metric but not significant' when discussing simulations. Either remove this statement and assume spiking is not relevant, or increase the number of simulations.

We could not find the exact quote from the reviewer, and we believe that he intended to quote “spiking is better on all metrics, but without significant margins”. Indeed, spiking did not improve the fit significantly on perturbed trials, this is particularly true in comparison with the benefits of Dale’s law and local inhibition. As suggested by the reviewer, we rephrased the sentence from this quote and more generally the corresponding paragraphs in the intro (lines 83-87) and in the results (lines 245-271). Our corrections in the results sections are also intended to address the minor point (4) raised by the same reviewer.

(3) The authors prefer the metric of predicting hits over MSE, especially when looking at real data (Figure 3). I would bring the supplementary results into the main figures, as both metrics are very nicely complementary. Relatedly, why not add Pearson correlation or R2, and not just focus on MSE Loss?

In Figure 3 for the in-vivo data, we do not have simultaneous electrophysiological recordings and optogenetic stimulation in this dataset. The two are performed on different recording sessions. Therefore, we can only compare the effect of optogenetics on the behavior, and we cannot compute Pearson correlation or R2 of the perturbed network activity. To avoid ambiguity, we wrote “For the sessions of the in vivo dataset with optogenetic perturbation that we considered, only the behavior of an animal is recorded” on line 294.

(4) I really like the 'forward-looking' experiment in closed loop! But I felt that the relevance of micro perturbations is very unclear in the intro and results. This could be better motivated: why should an experimentalist care about this forward-looking experiment? Why exactly do we care about micro perturbation (e.g., in contrast to non-micro perturbation)? Relatedly, I would try to explain this in the intro without resorting to technical jargon like 'gradients'.

As suggested, we updated the last paragraph of the introduction (lines 88 - 95) to give better motivation for why algorithmically targeted acute spatio-temporal perturbations can be important to dissect the function of neural circuits. We also added citations to recent studies with targeted in vivo optogenetic stimulation. As far as we know the existing previous work targeted network stimulation mostly using linear models, while we used non-linear RNNs and their gradients.

Minor:(1) In the intro, the authors refer to 'the field' twice. Personally, I find this term odd. I would opt for something like 'in neuroscience'.

We implemented the suggested change: l.27 and l.30

(2) Line 45: When referring to previous work using data-constrained RNN models, Valente et al. is missing (though it is well cited later when discussing regularization through low-rank constraints)

We added the citation: l.45

(3) Line 11: Method should be methods (missing an 's').

We fixed the typo.

(4) In line 250, starting with 'So far', is a strange choice of presentation order. After interpreting the results for other biological ingredients, the authors introduce a new one. I would first introduce all ingredients and then interpret. It's telling that the authors jump back to 2B after discussing 2C.

We restructured the last two paragraphs of section 2.1, and we hope that the presentation order is now more logical.

(5) The black dots in Figure 3E are not explained, or at least I couldn't find an explanation.

We added an explanation in the caption of Figure 3E.

**Reviewer #2 (Public review):**
(1) Some aspects of the methods are unclear. For comparisons between recurrent networks trained from randomly initialized weights, I would expect that many initializations were made for each model variant to be compared, and that the performance characteristics are constructed by aggregating over networks trained from multiple random initializations. I could not tell from the methods whether this was done or how many models were aggregated.

The expectation of the reviewer is correct, we trained multiple models with different random seeds (affecting both the weight initialization and the noise of our model) for each variant and aggregated the results. We have now clarified this in Methods 4.6. lines 658-662.

(2) It is possible that including perturbation trials in the training sets would improve model performance across conditions, including held-out (untrained) perturbations (for instance, to units that had not been perturbed during training). It could be noted that if perturbations are available, their use may alleviate some of the design decisions that are evaluated here.

In general, we agree with the reviewer that including perturbation trials in the training set would likely improve model performance across conditions. One practical limitation explaining partially why we did not do it with our dataset is the small quantity of perturbed trials for each targeted cortical area: the number of trials with light perturbations is too scarce to robustly train and test our models.

More profoundly, to test hard generalizations to perturbations (aka perturbation testing), it will always be necessary that the perturbations are not trivially represented in the training data. Including perturbation trials during training would compromise our main finding: some biological model constraints improve the generalization to perturbation. To test this claim, it was necessary to keep the perturbations out of the training data.

We agree that including all available data of perturbed and non-perturbed recordings would be useful to build the best generalist predictive system. It could help, for instance, for closed-loop circuit control as we studied in Figure 5. Yet, there too, it will be important for the scientific validation process to always keep some causal perturbations of interest out of the training set. This is necessary to fairly measure the real generalization capability of any model. Importantly, this is why we think out-of-distribution “perturbation testing” is likely to have a recurring impact in the years to come, even beyond the case of optogenetic inactivation studied in detail in our paper.

**Recommendation for the authors:**

**Reviewer #1 (Recommendation for the authors):**
The code is not very easy to follow. I know this is a lot to ask, but maybe make clear where the code is to train the different models, which I think is a great contribution of this work? I predict that many readers will want to use the code and so this will improve the impact of this work.

We updated the code to make it easier to train a model from scratch.

**Reviewer #2 (Recommendation for the authors):**
The figures are really tough to read. Some of that small font should be sized up, and it's tough to tell in the posted paper what's happening in Figure 2B.

We updated Figures 1 and 2 significantly, in part to increase their readability. We also implemented the "Superficialities" suggestions.